# Antimicrobial Resistance: A Growing Serious Threat for Global Public Health

**DOI:** 10.3390/healthcare11131946

**Published:** 2023-07-05

**Authors:** Md. Abdus Salam, Md. Yusuf Al-Amin, Moushumi Tabassoom Salam, Jogendra Singh Pawar, Naseem Akhter, Ali A. Rabaan, Mohammed A. A. Alqumber

**Affiliations:** 1Department of Basic Medical Sciences, Kulliyyah of Medicine, International Islamic University Malaysia, Kuantan 25200, Malaysia; 2Purdue University Interdisciplinary Life Sciences Graduate Program, Purdue University, West Lafayette, IN 47907, USA; amin50@purdue.edu; 3Department of Chemistry, Purdue University, West Lafayette, IN 47907, USA; 4Department of Pharmacy, University of Rajshahi, Rajshahi 6205, Bangladesh; moushumi5350@gmail.com; 5Department of Medicinal Chemistry and Molecular Pharmacology, Purdue University, West Lafayette, IN 47907, USA; pawar.72@osu.edu; 6The Ohio State University Comprehensive Cancer Center, Arthur G. James Cancer Hospital and Richard J. Solove Research Institute, Columbus, OH 43210, USA; 7Department of Neurology, Henry Ford Health System, Detroit, MI 48202, USA; nakhter2@hfhs.org; 8Molecular Diagnostic Laboratory, Johns Hopkins Aramco Healthcare, Dhahran 31311, Saudi Arabia; arabaan@gmail.com; 9College of Medicine, Alfaisal University, Riyadh 11533, Saudi Arabia; 10Laboratory Medicine Department, Faculty of Applied Medical Sciences, Albaha University, Al Baha 65431, Saudi Arabia; maali@bu.edu.sa

**Keywords:** antibiotics, antimicrobial resistance, mechanisms of resistance, drivers of resistance, measures to combat resistance

## Abstract

Antibiotics are among the most important discoveries of the 20th century, having saved millions of lives from infectious diseases. Microbes have developed acquired antimicrobial resistance (AMR) to many drugs due to high selection pressure from increasing use and misuse of antibiotics over the years. The transmission and acquisition of AMR occur primarily via a human–human interface both within and outside of healthcare facilities. A huge number of interdependent factors related to healthcare and agriculture govern the development of AMR through various drug-resistance mechanisms. The emergence and spread of AMR from the unrestricted use of antimicrobials in livestock feed has been a major contributing factor. The prevalence of antimicrobial-resistant bacteria has attained an incongruous level worldwide and threatens global public health as a silent pandemic, necessitating urgent intervention. Therapeutic options of infections caused by antimicrobial-resistant bacteria are limited, resulting in significant morbidity and mortality with high financial impact. The paucity in discovery and supply of new novel antimicrobials to treat life-threatening infections by resistant pathogens stands in sharp contrast to demand. Immediate interventions to contain AMR include surveillance and monitoring, minimizing over-the-counter antibiotics and antibiotics in food animals, access to quality and affordable medicines, vaccines and diagnostics, and enforcement of legislation. An orchestrated collaborative action within and between multiple national and international organizations is required urgently, otherwise, a postantibiotic era can be a more real possibility than an apocalyptic fantasy for the 21st century. This narrative review highlights on this basis, mechanisms and factors in microbial resistance, and key strategies to combat antimicrobial resistance.

## 1. Introduction

Antibiotics are the “magic bullets” for fighting against bacteria and are considered as the most remarkable medical discovery of the 20th century. The introduction of antibiotics has changed the therapeutic paradigm and continues to save millions of lives from bacterial infections. Antibiotics have absolutely been a godsend to humankind; they do not just have medicinal uses, but they have also been exploited in diverse purposes including animal husbandry and animal production as preventive measures in many underdeveloped and developing countries for decades [1]. With their ever-increasing use and misuse, microorganisms have developed antimicrobial resistance (AMR). The phenomenon of antimicrobial resistance refers to the potential of microorganisms including bacteria, viruses, fungi, and parasites to thrive and continue to grow in the midst of drugs designed to kill them. Infections caused by antimicrobial-resistant organisms are not only difficult to treat, there is also always an increased chance of severe illness and even death due to these infections. There are several types of antimicrobial agents including antibiotics, antifungal, antiviral, disinfectants, and food preservatives that either suppress the growth and multiplication of microbes or kill them. Antibiotics are a class of antimicrobials specifically used to combat bacterial infections and antibiotic resistance that is much more frequently used than any other class of antimicrobials. AMR is an unavoidable evolutionary phenomenon shown by all organisms through development of genetic mutations in order to safeguard the lethal selection pressure. To withstand the environmental selection pressure, bacteria strive to develop resistance against antibacterial drugs, rendering these drugs ineffective [2]. With the ever-increasing use of antibiotics, especially in developing countries, bacteria have ample opportunity to develop AMR with profound consequences including much higher morbidity and mortality [3,4,5]. The incidence and prevalence of antimicrobial-resistant-bacterial infections has attained incongruous levels during 21st century and threatens global public health as a silent pandemic, necessitating urgent interventions [6]. Antibiotic resistance can happen in any country and can affect anyone irrespective of age and gender. With its current scenario, AMR is one of the unsurpassed threats not only to global health but also to food security [7]. Evolution and dissemination of AMR is concurrently affected by a huge number of interdependent factors related to healthcare and agriculture. In addition, it can also be affected by factors contributing from pharmaceuticals, inappropriate waste management, trade, and finance, creating AMR as one of the most intricate public health concerns worldwide [8]. With the rapid global spread of “superbugs” (microorganisms that are resistant to most known antimicrobials), the situation of drug-resistant pathogens has attained a real and alarming status. AMR has been acknowledged to be one among the top three major public health threats by the World Health Organization (WHO). Antimicrobial-resistant infection has been ranked third as the leading cause of death after cardiovascular diseases [9]. An estimated 1.27 million deaths were attributable to antimicrobial-resistant infections in 2019 alone, while nearly 5 million deaths were somehow associated with drug-resistant infections, according to a major study published in January 2022. This number is estimated to be increased to 10,000,000 per year by 2050, greatly exceeding deaths from cancer [10]. As a well-known example of first “superbug”, methicillin-resistant *Staphylococcus aureus* (MRSA) is associated with a high death toll from antimicrobial-resistant infections across the globe [11]. Currently, 3.5% of active TB and 18% of previously treated TB cases belong to MDR-TB (multidrug-resistant tuberculosis) worldwide and there is a growing concern for XDR-TB (extensively drug-resistant tuberculosis) among many MDR-TB cases [12]. Although antibiotics are essential in combating bacterial infections, their misuse and abuse with inappropriate dose and duration over the decades have resulted in selection pressure with the emergence of resistant bacteria. Apart from human healthcare, the emergence and spread of AMR from the unscientific use of antimicrobials in livestock feed in many developing countries has been a major contributing factor. It necessitates increased surveillance on the impact of excessive and unregulated use of antibiotics in animal feeds to downturn the incidence of drug-resistant bacteria [13]. Antibiotic resistance can have an impact on human health in terms of both therapeutic and preventive consequences. The therapeutic implication is direct and seen through treatment failure and complications, while preventive implications are seen through compromise of treatment options for immunosuppressive situations such as cancer chemotherapy, advanced surgical procedures such as transplantation, and invasive procedures such as intubation or catheterization [14,15]. The current investment in the development of new synthetic small and natural-product-derived molecules stand in sharp contrast with an ever-growing demand of novel antimicrobials to treat life-threatening antimicrobial-resistant infections. Pharmaceutical giants have abandoned their interest in antibiotic discovery, based on their own reasoning and have ceased increasing their significant antibiotics inventory since 1980s. Fluoroquinolone was added in the group of the last broad-spectrum antibiotics discovery in the 1980s and was brought to market in 1987. Since then, there has been a paucity in the development and only a few new antibiotic groups are in the pipeline [16]. Use of antibiotics is intertwined with the development of resistance, implying that resistance can be substantially reduced by avoiding unnecessary consumption of antibiotics. Given the fact that antimicrobials are indispensable tools to treat and prevent infectious diseases, it is now crucial to preserve the efficacy of currently available antimicrobials since there has been no significant discovery of new molecules during recent decades [17]. This narrative review highlights the basis, mechanisms, and factors of microbial resistance and key strategies to combat antimicrobial resistance. 

## 2. Timeline of Major Antibiotics Discoveries and Antibiotic Resistance 

The dawn of the modern antibiotic era can be marked by the discovery of salvarsan and neosalvarsan, a synthetic prodrug, by Paul Ehrlich in 1910 to treat syphilis caused by *Treponema pallidum.* Later on, prontosil, a sulfonamide prodrug discovered by bacteriologist Gerhard Domagk, gradually replaced salvarsan. Selman Waksman, an American microbiologist and biochemist, is credited with the first systematic evaluation of microbes in soil and their ability to generate compounds with antimicrobial action in the 1930s. He unearthed multiple antibiotics from filamentous actinomycetes living in the soil including streptomycin, a widely used antibiotic against tuberculosis, and defined an antibiotic as “*a compound made by a microbe to destroy other microbes*”. Penicillin was discovered from a mold called *Penicillium rubens* by Sir Alexander Fleming, a Scottish physician and microbiologist, in 1928, which began the golden era of antibiotic discovery that peaked until mid-1950s. With regard to antibiotic discovery, the period from the 1940s to the 1960s is regarded as the “Golden Age”, and most of the antibiotics still in current use were discovered during that period. Since then, there has been a gradual decline in antibiotic discovery with concomitant evolution of drug-resistant pathogens. Bacterial resistance to antibiotics has been recognized almost since the dawn of the antibiotic era [16]. Several years before the introduction of penicillin as a therapeutic agent in 1940, the first penicillin-resistant *Staphylococcus* strain was described. Methicillin was introduced in 1959 as the first semisynthetic penicillinase-resistant penicillin, and surprisingly, a methicillin-resistant *Staphylococcus* strain was reported in 1960, just a year after its introduction [18]. In 1958, vancomycin, a glycopeptide, was introduced as a rescue drug for treating infections caused by methicillin-resistant *Staphylococci*, but unfortunately in 1979, a couple of decades later, vancomycin-resistant strains of coagulase-negative *Staphylococci* (CoNS) were reported, and ten years later vancomycin-resistant *Enterococcus* (VRE) was also described. Decreased efficacy of vancomycin was subsequently noted for *S. aureus*, with vancomycin-intermediate *Staphylococcus aureus* (VISA) and vancomycin-resistant *Staphylococcus aureus* (VRSA) reported in 1997 and 2002, respectively [19]. Cephalosporin, a β-lactam antibiotic, was discovered in 1945 and introduced in clinical practice in 1964 to treat penicillin-resistant cases; since then, several generations of cephalosporins have been launched, with the fifth generation being currently available. It was excellent in its efficacy to start with, especially against extended beta-lactamases (ESBLs) producing gram-negative bacteria. Until recently, all previous generations of cephalosporin up to the fourth generation have developed significant resistance. Tetracycline is another important antibiotic, discovered in 1950, and was successfully used for many common infections including gastrointestinal diseases. Within a decade of its discovery, tetracycline was reported to be inefficacious to *Shigella* strains in 1959. Levofloxacin, a member of the third-generation fluoroquinolone, was added to the antibiotics list in 1996, and levofloxacin-resistant *Pneumococcus* was reported in the same year [20]. Carbapenem is a type of β-lactam that was introduced in 1980 and preserved to be a reserve drug to treat infections caused by members of enterobacterales, especially cephalosporin-resistant cases. With its increased use during the period of the 1990s to 2000s, carbapenem-resistant enterobacterales (CRE) have been reported from different countries since 2006 [21]. From the timeline of antibiotic discovery, it is evident that new classes of antibiotics were produced by the pharmaceutical industries only for two decades, i.e., from 1960 to 1980, and afterwards there was a dramatic decrease in the speed of discovery until recently [17]. This disproportionate ratio between drug-resistant pathogens and number of available antibiotics has given sufficient reasons to critics for their prediction of an imminent postantibiotic era. The timeline of major antibiotics discovery and their resistances is depicted below (Figure 1).

## 3. Superbugs

Superbugs refer to germs that have shown resistance to antimicrobial agents used to treat them and include multidrug- or pan-drug-resistant bacteria and fungi. In reality, there is scarce or no treatment at all available for infections caused by superbugs. The term “*ESKAPE*” is the acronym for six highly drug-resistant bacteria (*Enterobacterales*, *Staphylococcus aureus*, *Klebsiella pneumoniae*, *Acinetobacter baumannii, Pseudomonas aeruginosa*, and *Enterobacter*) and at present, carbapenem-resistant enterobacterales (CRE), carbapenem-resistant *Klebsiella pneumoniae* (CRKP), methicillin-resistant *Staphylococcus aureus* (MRSA), ESBL-producing enterobacterales, vancomycin-resistant *Enterococcus* (VRE), multidrug-resistant *Pseudomonas aeruginosa*, and multidrug-resistant *Acinetobacter* are among the topmost encountered superbugs worldwide. Multidrug-resistant bacteria have emerged only after long-continued and widespread use of antibiotics to treat infections caused by them. For example, *M. tuberculosis* has turned out as MDR-TB after decades of treatment with antitubercular drugs, now found as a major superbug prevalent in both underdeveloped and developing countries. Hospital-acquired or healthcare-associated infections (HAIs) caused by both gram-positive (e.g., *Staphylococcus epidermidis*, *Clostridioides difficile*, and *Streptococcus pneumoniae*) and gram-negative (e.g., *Burkholderia cepacia*, *Stenotrophomonas maltophilia*, *Campylobacter jejuni*, *Citrobacter freundii*, *Enterobacter* spp., *Haemophilus influenzae*, *Proteus mirabilis*, *Salmonella* spp., and *Serratia* spp.) bacteria are considered as superbugs because most of the available antibiotics have been proven ineffective to treat them [22]. Infections with superbugs enhance the rate of morbidity and mortality, as therapeutic options for these bacteria are seriously jeopardized and also there are high treatment costs and extended periods of hospital stay associated with these infections [23].

## 4. Basis of Antibiotic Resistance 

Antibiotic resistance is an evolutionary response of bacteria that develops against the challenge of therapeutic antibiotics. From a clinical perspective, all targeted pathogens remain susceptible to an antibiotic when it is first launched, but with sustained use, bacteria develop resistance to it. From an evolutionary perspective, bacteria adapt the action of antibiotics by either (1) chromosomal gene mutations, or (2) acquisition of foreign DNA through horizontal gene transfer (HGT) that codes for resistance determinants. Mutations principally involve three different types of genes, viz., genes encoding the targets of the antibiotic, transporters of the antibiotic, and regulators that repress the expression of transporters (e.g., antibiotic-modifying enzymes and multidrug efflux pumps) to give rise to antibiotic resistance. There is intriguing evidence to support the notion that commensal or environmental bacteria are the source of the antibiotic-resistance gene(s) that are transmitted to human pathogenic bacteria through HGT [24]. It is well-known that there are many antibiotics naturally synthesized by environmental microorganisms. To safeguard them from the action of self-synthesized antibiotics, they must possess antibiotic-resistant genes too, otherwise they would have been killed by their own antibiotics [25]. 

Bacteria exhibiting antibiotic resistance can have gene(s) from intrinsic, acquired, or adaptive sources [26].

*Intrinsic resistance* refers to bacteria’s inherent natural capacity to show resistance to certain classes of antibiotics due to the presence of their own chromosomal genes without mutation or gain of further genes. The implication of intrinsic resistance is that these bacteria will show inevitable resistance against certain antibiotics if used to treat infections by them. As far as the drug-resistance mechanisms are concerned, both efflux pumps and reduced permeability are involved in intrinsic resistance. It can also affect the multidrug efflux pumps frequently [27,28].

*Acquired resistance* is defined as an evolutionary process of exhibiting the resistance by a previously sensitive bacterium due to acquisition of chromosomal gene mutation or gaining an exogenous new genetic material via HGT. There are three main mechanisms for HGT, viz., transformation, transposition, and conjugation. The acquired resistance is most often transmitted through a plasmid acquired via conjugation, and it may be temporary or permanent [29,30]. 

*Adaptive resistance* is a phenotype that is conditional to environmental changes, and depending on the ability and duration of selection pressure, it may be interim or permanent. When bacterial growth is influenced by subinhibitory concentrations of antibiotics along with specific environmental signals such as growth factors, nutrition, stress, pH, concentrations of ions, etc., bacteria can develop adaptive resistance in both humans and livestock. As opposed to intrinsic and acquired resistance phenotypes, adaptive resistance is usually developed transiently and generally reverts back to the original state upon removal of the inducing signals. Although the exact biological processes involved in the evolution of adaptive resistance are not well understood, several factors including high mutation rates, gene amplification, efflux pumps, biofilm formation, epigenetic inheritance, population structure, and heterogeneity have been mentioned as possible explanations for its development [31,32]. 

## 5. Sources and Routes of Transmission of AMR

The transmission and acquisition of AMR occur primarily via the human–human interface both within and outside of healthcare facilities. Humans, animals, water, and the environment are found to be reservoirs, and antimicrobial-resistance genes can be transmitted between and within these reservoirs. As far as the transmission routes are concerned, there is significant difference between bacterial species and resistance elements [33]. 

Transmission of antimicrobial-resistant bacteria is much facilitated by certain hotspot sources such as wastewater and sludge from urban wastewater treatment plants, and natural fertilizers such as pig slurry, cow manure, and fertilizer from poultry farming [34]. Animal feeds treated with antibiotics and their subsequent transfer to humans through consumption of these animals constitute the direct route of acquisition of antimicrobial resistance from animals [35]. Further, ingestion of fecal-contaminated food or water and direct contact between animals and humans constitute other common routes of transmission [36]. 

## 6. Mechanisms of Drug Resistance

Antimicrobials and bacteria coexist in the same ecological niche, and bacteria develop defenses against the harmful effects of antibiotic molecules. There are four essential targets in a bacterial cell for antibiotics (e.g., cell wall, cell membrane, protein synthesis, and nucleic acid synthesis). Primary mechanisms for antimicrobial resistance include limiting drug uptake, altering a drug target, inactivating a drug, and increasing active drug efflux (Figure 2).

In general, for acquired resistance, bacteria use mechanisms such as modification of drug target, drug inactivation, and drug efflux, whereas intrinsic resistance mostly results from restricting uptake, drug inactivation, and drug efflux. Gram-positive and gram-negative bacteria differ in their structural makeup, which causes variance in their drug-resistance mechanisms. Gram-positive bacteria less frequently utilize the method of restricting uptake of a drug because they lack the lipopolysaccharide (LPS) outer membrane and have limited capacity for an efflux mechanism to certain types of drugs [37,38]. Meanwhile, gram-negative bacteria have been shown to use all four main mechanisms of drug resistance. 

### 6.1. Limiting Drug Uptake

Lipopolysaccharide, a highly acylated glycolipid, forms the major component of the outer membrane of gram-negative bacteria and serves as a permeability barrier for a variety of chemicals, including antibiotics. This intrinsic resistance of gram-negative bacteria lowers permeability of specific antibiotics, leading to resistance. Additionally, modifications in the permeability of outer-membrane proteins, in particular porin protein, can lead to acquired drug resistance. Porins serve as the primary entry point for hydrophilic antibiotics such as β-lactams, fluoroquinolones, tetracyclines, and chloramphenicol. The quantity and type of porin proteins have an impact on the entry of these antibiotics into the bacterial cell and consequently affect bacterial susceptibility to these antibiotics [39]. Furthermore, acquired antibiotic resistance may result from mutations that impair the expression of porin or its function. Mutations affecting the expression of porin lead to high levels of resistance when combined with other co-existing mechanisms, such as efflux pumps or enzymatic degradation of antibiotics [40]. The formation of biofilms by some bacteria (e.g., *Enterococcus faecalis*, *Staphylococcus aureus*, *Staphylococcus epidermidis*, *Streptococcus viridans*, *E. coli*, *Klebsiella pneumoniae*, *Proteus mirabilis*, and *Pseudomonas aeruginosa*) is another method of antimicrobial resistance shown by these bacteria. A biofilm is an assemblage of microbial cells embedded in self-produced exopolysaccharide and attached to abiotic or biotic surfaces. It is known to confer bacterial tolerance and resistance to antibiotics through a variety of processes including obstruction of antibiotic penetration. Further, it may stop the building of bactericidal concentrations of antibiotics over the entire biofilm [41,42]. 

### 6.2. Modification of Targets for Drug 

Bacteria can modify the targets required for drug binding so that the drug cannot bind or binds poorly to the modified target. This modification results from spontaneous mutations of the gene or genes that encode the protein that acts as the drug target. For instance, when mutations impact the quinolone-resistance-determining region (QRDR) in the DNA gyrase (topoisomerase II and topoisomerase IV), fluoroquinolone resistance develops in both gram-positive and gram-negative bacteria [43]. Another way of target modification is methylation, which is considered to be a very efficient method in developing resistance. Examples of methylation include *erm* methylases against macrolides, lincosamides, and streptogramin B antibiotics in both gram-positive and gram-negative bacteria. Similarly, methylation of the *cfr* gene has been linked to the development of resistance in a variety of bacteria, including *Proteus vulgaris*, *Staphylococcus* spp., *Enterococcus* spp., *Bacillus* spp., and *E. coli* [44]. *Staphylococcus* spp. exhibits a significant reduction in its affinity to β-lactam antibiotics due to an alternative penicillin-binding protein encoded by *mecA* and *mecC* genes [45,46].

### 6.3. Inactivation of Drug 

Drug resistance may result from the inactivation of antibiotics by certain bacterial species that follows in one of two ways: either the antibiotic is really degraded, or a chemical group is transferred to the antibiotic. The hydrolyzing enzymes known as β-lactamases, produced by members of the enterobacterales family, are particularly effective at inactivating β-lactam antibiotics. The β-lactamases originally known as penicillinases and cephalosporinases inactivate the β-lactam ring structure by opening at a specific point rendering it ineffective to bind with the target, called penicillin-binding proteins. Several members of the enterobacterales family as well as many species of gram-positive bacteria such as *Staphylococcus aureus*, *Enterococcus faecalis*, and *Enterococcus faecium*, are known to harbor β-lactamase genes that are transmitted by HGT. Additionally, tetracycline is hydrolyzed by an enzyme, expressed by the *tetX* gene present in certain bacteria [47]. The transfer of acetyl, phosphoryl, and adenyl groups is the most frequently seen chemical groups for inactivation of drugs. Phosphorylation and adenylation are known to be utilized predominantly against aminoglycosides, while acetylation is the most diversely used mechanism against aminoglycosides, chloramphenicol, streptogramins, and fluoroquinolones [38].

### 6.4. Efflux of Drug 

By using an energy-dependent efflux pump located on the cytoplasmic membrane, bacteria are able to control the accumulation of antibacterial chemicals, including antibiotics, inside bacterial cells. By expelling harmful compounds such as antibiotics, metabolites, and quorum-sensing signal molecules from the cell, efflux pumps enable bacteria to control their internal environment. In 1980, researchers described the first plasmid-encoded efflux pump in *Escherichia coli*, which pushed tetracycline out of the bacterial cell. Since then, numerous gram-positive and gram-negative resistant bacteria with diverse efflux mechanisms have been found. It is interesting to note that the majority of efflux systems engage in multidrug efflux mechanisms that are always chromosomally encoded to ensure bacterial intrinsic drug resistance [48]. Instead, genes for substrate-specific efflux pumps (for example those for chloramphenicol, tetracyclines, and macrolides) are more likely to be found on mobile genetic elements [38,49].

There are six drug efflux pumps based on the structure and energy source, namely, the ATP-binding cassette (ABC) superfamily, the major facilitator superfamily (MFS), the multidrug and toxic compound extrusion (MATE) family, the small multidrug resistance (SMR) family, the resistance–nodulation–division (RND) superfamily, and the drug metabolite transporter (DMT) superfamily. The majority of the efflux pumps that are present in gram-positive bacteria belong to the ABC and MFS families and are either carried on plasmids or encoded by chromosomal genes, while the main clinically significant efflux systems in gram-negative bacteria are members of the RND superfamily, which is typically made up of an outer-membrane protein channel, a periplasmic protein, and a cytoplasmic membrane pump [50].

## 7. Drivers to AMR

Antimicrobial resistance is driven by multifaceted drivers including inherent traits of the microbes and many environmental factors that involve both prescribers and consumers. Broadly, factors contributing AMR can be of four categories, including environmental factors (e.g., population and overcrowding, rapid spread through mass travelling, poor sanitation, ineffective infection control program, and widespread agricultural use), drug related factors (e.g., fake drugs, substandard drugs, and over-the-counter availability), patient-related factors (poor compliance, poverty, lack of education, self-medication, and misconception) and physician-related factors (inappropriate prescription, inadequate dosing, and lack of updated knowledge and training) [38,51]. Some of the recognized AMR driving factors are elaborated below.

### 7.1. Misuse and Overuse of Antibiotics

Although the process of development of antibiotic resistance occurs as a natural phenomenon, it has been accelerated by the misuse of antibiotics in both humans and animals over the years. There is a causal relationship between overuse and development of microbial resistance to antibiotics as revealed by epidemiological studies [52]. Despite being warned repeatedly by health organizations, unfortunately, misuse and overuse of antibiotics continue at a disproportionate level worldwide, and the current scenario seems to be at the point of no return. 

Surveys have revealed that people across the globe, especially noneducated sections, do have misconceptions and beliefs about antibiotics, for example, that they help to recover from most common viral diseases, such as the common cold or flu. Moreover, it has been observed that antibiotics are a frequently prescribed medicine for patient management, particularly observed in many developing countries where there is lack of adequate diagnostic facilities [53]. Administering antibiotics without a clear indication is a good example of common misuse. The emergence and spread of drug-resistant pathogens are facilitated further when antibiotics can be bought for human as well as animal use as over-the-counter (OTC) drugs. Antibiotic abuse is also contributed to by lack of antibiotic policy and standard treatment guidelines, frequently seen in developing countries. Further, antibiotics are often overprescribed by health workers, pharmacy dealers, and veterinarians in many underdeveloped and developing countries. Substandard or poor quality antibiotics in the supply chain has made the situation of AMR worst in many developing countries. Moreover, antibacterial resistance can also be contributed to by physicians unnecessarily prescribing lengthy courses of antibiotics or dosing inappropriately. Unethical though it is, sometimes to gain financial incentives from pharma companies and to satisfy patients’ expectations, many physicians especially in the developing countries prescribe antibiotics without indication [53,54].

### 7.2. Increase in Gross Domestic Product (GDP) 

The significant rise in antibiotic use globally is predominantly accredited to the rise in the GDP especially in many developing countries. With the rise in GDP, there has been substantial improvement in the quality of life of people from low- and middle-income countries (LMICs) that positively correlates with increased antibiotic consumption. It is estimated that between 2000 and 2015, global antibiotic use has been elevated by 65%, according to Klein et al. [55]. Further, with the rise in GDP, consumption of animal protein has also risen in many developing countries, which further adds to the transmission of AMR from animal sources in these countries [56].

### 7.3. Inappropriate Prescribing Patterns 

Inappropriately prescribed antibiotics contribute significantly to promoting AMR [57]. Inappropriate antibiotic prescribing refers to prescription of antibiotics where it is not necessary or selection of inappropriate antibiotics or the wrong dose and duration of an antibiotic [58]. It has been shown in a study that at least one antibiotic was received by 50% of patients without compelling reasons during their stay at the hospital. The introduction of antibiotics should ideally be guided by prior isolation and antimicrobial susceptibility testing of bacteria, but according to a CDC (Centers for Disease Control and Prevention) report from 2017, antibiotic prescriptions were made for about one-third of hospital patients without adequate testing and continued for longer durations [59]. Situations in nursing homes are even worse, where about 75% of the antibiotic prescriptions are incorrect or inappropriate, with wrong doses and durations [60]. 

### 7.4. Paucity in Futuristic Antibiotics 

The looming problem of antibiotic resistance demands urgent response by the pharma companies with new novel antibiotics [61]. Unfortunately, there is a dearth of development in new antibiotics, despite having repeated calls from the WHO. Surprisingly, only 8 out of the 51 newly developed antibiotics can be catalogued as innovative drugs to treat infections caused by antibiotic-resistant bacteria; the overwhelming majority are just reformations of previous drugs. As a consequence, it is speculated that these new drugs are likely to show resistance shortly. The current scenario states that management of drug-resistant TB, urinary tract infections, pneumonia, and other gram-negative infections has been seriously jeopardized because of lack of available treatment options. The paucity of new drugs has made patients of extreme of age much more vulnerable to life-threatening infections [62]. Regulatory restrictions and economic liability are major hindrances to production of new antibiotics, according to pharmaceutical companies. On that account, many organizations have reduced their investment in the research and innovation category substantially, and surprisingly, 18 major companies have abandoned their antibiotics production. Considering profit, pharma companies have shifted their priority and are now interested in producing drugs more for chronic diseases than infectious diseases. 

### 7.5. Agricultural Use of Antibiotics

Use of antibiotics in livestock farming has been markedly increased in most developing countries for various purposes, including increased demand of animal protein in recent years. In turn, it is contributing to AMR due to the presence of antibiotic residues in animal-derived products (e.g., muscles, kidney, liver, fat, milk, and egg). Antibiotics are being used randomly for various purposes including treating animal diseases, preparation of animal feed for growth promotion, improved feed conversion efficiency, as well as for disease prevention [63]. This practice is more prevalent in developing countries to gain more income from the food animal farms and due to lack of regulatory government policies, which has been a major contributor to human AMR [64]. Approximately 70% of all medically important antibiotics are sold for use in animals in the United States [65]. It is a matter of great concern that the uses, types, and modes of action of antibiotics employed in veterinary practice are closely related to or the same as those prescribed to humans. 

### 7.6. Easy Travel Routes

There is growing evidence that the emergence and global spread of antibiotic-resistant bacteria are much facilitated by human movement. Dissemination of AMR across the globe is contributed to significantly by easy and modern travelling routes available not only for humans but also for animals and goods [66]. Human travellers are highly likely to return to their own countries, unknowingly with colonization or infection by antimicrobial-resistant organisms, from the countries they have visited. It has been shown that antimicrobial-resistant bacteria may persist for up to 12 months carried in the body after a person has travelled to highly endemic AMR regions, amplifying the risk of transmission among susceptible populations [67].

### 7.7. Knowledge Gap

There is substantial evidence that both healthcare workers (HCWs) and members of the public have knowledge gaps about appropriate use of antibiotics and mechanisms of antibiotic resistance [68]. Surveillance is a prerequisite to estimate the magnitude of the AMR burden and to establish any intervention strategies, such as antimicrobial stewardship. Unfortunately, actual statistical data regarding the use of antibiotics and the status of AMR in both healthcare and agriculture sectors have yet to be gathered worldwide [69]. Surveillance data provide critical information and help in identifying areas for strategic interventions to maximize the outcomes. In order to launch successful intervention approaches through cooperative efforts from different stakeholders (e.g., international agencies, human and veterinary medicine sectors, agriculture and animal production industries, and consumers), the existing knowledge gap needs to be addressed first. 

## 8. Clinical Implications of AMR

There are many clinical implications of AMR, and following are some of the major concerns [70]:Successful treatment of microbial infections including bacterial, fungal, and viral infections is hindered by antimicrobial resistance.Emergence and dissemination of new resistant mechanisms threaten the scope of treatment for many common illnesses such as urinary tract infections, upper respiratory tract infections, typhoid, and flu, resulting in treatment failure, permanent disability, or even death.The success of cancer chemotherapy, transplantation surgery, and even minor dental procedures will be seriously jeopardized by virtue of AMR unless novel drugs are available.AMR infections impose mandatory prolonged treatment with higher healthcare costs and may require expensive alternate drugs.

## 9. How to Combat AMR

Antimicrobial resistance is a serious concern affecting not only people but animals, plants, and the environment at large. Like humans, sometimes animals can be a potential source of MDR germs, which can be transmitted through close contact or consumption of foods from animal sources. No single government department or independent organization in any country can tackle the problem of ever-growing AMR alone. To contain and control AMR, it demands an orchestrated coordination and collaboration within and between multiple sectors, such as healthcare industries including pharmacy, agriculture, finance, trade, education, and nongovernment organizations at national and international levels. Multisectoral collaboration can be both horizontal and vertical. Horizontal collaboration is across sectors and departments within the country, i.e., multistakeholder forums, and vertical collaboration involves different levels within a country, a region, and internationally [71]. 

The trend of physicians prescribing broad-spectrum antibiotics for trivial conditions needs to be checked immediately, and the usage of antimicrobials for animals by veterinarians also needs close monitoring. In order to combat AMR, rational antibiotic prescription, limited use of prophylactic antimicrobials, patients’ education, compliance with antibiotic therapy, and appropriate hospital hygiene through antimicrobial stewardship are among the main focuses [72]. Furthermore, development and availability of faster diagnostic tools and accurate antimicrobial profiling for targeted antibiotic therapy are also important. 

The World Health Assembly adopted five key strategic action plans to combat AMR which include the following steps: (1) improve awareness and understanding of antimicrobial resistance; (2) strengthen knowledge through surveillance and research in combating infection through control measures; (3) implement effective sanitation, hygiene, and infection prevention measures; (4) optimize the use of antimicrobials in human and animal health; and (5) encourage sustainable investment in new medicines, diagnostic tools, and vaccines [73]. Highlights of a few important national and international measures to combat AMR are described below [69] and depicted in Figure 3.

### 9.1. International Measures

The following are international measures that can be taken:Establishing and strengthening collaboration among international agencies, governments, nongovernmental organizations, and professional groups.Establishing surveillance networks for antimicrobial use and AMR globally.Building laboratory capacity for the detection and reporting of pathogens with AMR that have global health impacts.Establishing and strengthening international tracking systems for quick identification and mitigation of emerging pathogens.International monitoring to control counterfeit antimicrobials across the globe.Investing in research, new drug discovery, and vaccines.

### 9.2. National Strategies

The following are national measures that can be taken:Implementing an “Antibiotic policy” for judicious use in healthcare and agricultural settings.Strengthening of national surveillance, monitoring, and evaluation efforts by integration of public health and veterinary sectors.Developing innovative point-of-care diagnostic tests for pathogen identification and resistance monitoring.Investing in basic and applied research on new antibiotics and vaccines.Building capacity and strengthening international collaboration to combat AMR.Adopting antimicrobial stewardship in healthcare settings with essential drug list.

### 9.3. Rational Use of Antibiotics 

“Rational use of medicine” has been defined by the WHO as using correct medications including antibiotics appropriate for clinical needs of patients, in exact doses of individual needs, for an adequate period of time, and at the lowest cost [70]. The optimal results of treating infections can only be achieved when the selection of pathogens, drug toxicity, and development of resistance are minimized through the rational use of antibiotics. Antibiotic stewardship programs (ASPs) in healthcare settings are primarily aimed at maintaining the rational use of antibiotics. 

### 9.4. Ban on Over-the-Counter (OTC) Antibiotics

Stringent regulatory control should be imposed on OTC selling and dispensing of oral and injectable antibiotics, which is unfortunately still a common practice in many underdeveloped and developing countries. Antibiotics should only be dispensed to serve the prescription from a qualified physician. Continuous awareness programs on the use of antibiotics and AMR among patients and pharmacy drug dispensers is strongly recommended, along with reappraisal of existing antibiotic policies based on local and regional AMR surveillance data [74]. 

### 9.5. Infection Prevention and Control (IPC) 

Infection prevention and control (IPC) is an essential and evidence-based practical approach to safeguard both patients and healthcare workers from being victimized by avoidable infections including drug-resistant pathogens. This is an indispensable measure mitigating the AMR in healthcare settings. The role of physicians, nurses, pharmacists, and other healthcare providers are very crucial to combating AMR through IPC. The physicians involved in direct patient care can play a paramount role in combating AMR through complying with hospital infection control and antibiotic policies along with timely notification of resistant cases to the IPC team. In addition, nurses and other healthcare providers need to be educated about the AMR and aseptic practices in controlling the spread of infections. The role of the hospital pharmacist as an important member of the IPC team is to inculcate patients regarding treatment compliance including antimicrobial use, which contributes tremendously to combating AMR [75]. 

Recommended measures related to IPC in a healthcare facility include:Formation of “infection prevention and control committee”.Practices of good hand hygiene.Proper diagnosis and successful treatment of infection.Responsible use of antimicrobial agents.Continuous surveillance and monitoring of antibiotic use and antibiotic resistance.Establishing quality antimicrobials supply chain.Good microbiological laboratory practices.

### 9.6. Antimicrobial Stewardship Program (ASP)

Antimicrobial stewardship is a coordinated program to educate and persuade prescribers to follow the appropriate selection, dosage, and duration of antimicrobial agents for improved patient outcomes in order to reduce microbial resistance and its spread. The first goal of antimicrobial stewardship is to make sure that healthcare practitioners prescribe the most appropriate antimicrobial with the correct dose and duration for each patient. The second goal is pointed towards the prevention of overuse, misuse, and abuse of antimicrobials. The third goal is to keep resistance development to a minimum. There are two major overlapping approaches in achieving the primary goals of antimicrobial stewardship: (1) using antibiotics to optimize healthcare outcomes, and (2) using antibiotics to ensure sustainable access for all who need them. In 2014, the CDC released the “*Core Elements*” of antimicrobial stewardship in achieving these goals, which are applicable to all hospitals, regardless of size, with specific suggestions for small and critical-access hospitals in their implementation [76,77]. 

### 9.7. Use of Antibiotics in Animals

The WHO specifically called for stricter legislation in using medically important antibiotics in animals to curtail the problems of antimicrobial resistance. Further, it emphasizes on overall reduction and complete restriction of the use of antibiotics for the sake of growth promotion and disease prevention. Antibiotics to prevent disease can be administered if infection has been diagnosed in other animals in the same flock, herd, or fish population. As alternative measures, improved hygiene, provision of probiotics or nutritional supplements in feed, better use of vaccination, and changes in practices of animal husbandry are encouraged [78]. 

### 9.8. Development of New Drugs and Vaccines

Due to the rapid development of resistance to each new class of antibiotic and the challenges in producing new effective drugs, a focus on research into an integrated strategy that includes development of both vaccines and novel antibiotics is key. To combat AMR, greater investment is required in the operational research and innovation of new antimicrobials through collaborative efforts of academia and industries both at the national and international levels. Vaccines have been used as prophylactic measures to prevent infectious diseases for a long time and are considered an essential tool to decrease the demand for antimicrobial drugs and thereby combat AMR. Moreover, they are not blamed for resistance development like that of antibiotics. Thus, innovation and use of vaccines against antimicrobial-resistant bacterial infections, especially carbapenem-resistant enterobacterales and *Acinetobacter baumannii*, are of prime importance and could be a potential strategy to fight against AMR transmission [73,75].

### 9.9. Introduction of Checkpoints 

Practice of illegal sales and self-medication of antibiotics are still a prevailing trend observed in some countries, especially in underdeveloped and low-income countries where anyone can buy drugs from pharmacies without a prescription from a registered doctor. Sometimes the decision of antibiotic prescription by physicians is influenced by a patient’s desire, which is totally irresponsible. Stringent control and checkpoints should be introduced to contain these detrimental practices that escalate the development of AMR. Proper legislation and its implementation could be an appropriate step to limit the illegal sale of drugs, especially antibiotics. Other checkpoints may include delayed antibiotic prescribing (intake of antibiotic in a prescription is deferred until symptoms appear and the patient is clearly instructed on when to start an antibiotic), which is a successful strategy to combat AMR [58,60].

### 9.10. Community Engagement 

The development and propagation of AMR is often considered as a biological phenomenon that relates to many everyday practices of people in the community, such as home and animal hygiene, food production, health-seeking behaviors, and waste disposal. Moreover, it is likely that each community may have their own language and perception with regard to the use of antimicrobials and drug resistance. So, a community engagement approach towards behavioral change for antimicrobial use could potentially safeguard both existing and future treatment options and offer better strategies to combat AMR at the community level. It demands more research in engaging people to explore their understanding and experiences, with an aim to translating these ideas into applications to combat the AMR problem [79]. 

### 9.11. Alternatives to Antibiotics

Researchers are trying their best to find potential alternatives for antibiotics from natural resources. Plants are being considered as untapped sources of potential antimicrobial agents, according to recent findings on compounds derived from plants, such polyphenolics, alkaloids, and other plant extracts [80]. However, the antimicrobial potential of several phytochemical compounds including polyphenolics, alkaloids, and flavonoids has yet to be investigated [81,82,83,84,85,86]. Moreover, the big challenge is to transfer and translate some of these precious discoveries out of laboratories and into hospital practices. With the advent of advanced technological breakthroughs, especially in biotechnology, genetic engineering, and synthetic chemistry, opportunities for research on innovative alternative therapies have been widening, which brings optimism to the growing AMR problem. Scientists now know that microbes such as bacteria, viruses, and molds compete for resources; living next to each other, they are engaged in chemical warfare, i.e., to defend themselves from their own secreting chemicals. By manipulating the phenomenon of microbial chemical warfare, discoveries of new alternatives to antibiotics that attack disease-causing bacteria are on the way [87]. 

#### 9.11.1. Phage Therapy 

Phage therapy refers to the application of bacterial viruses to combat populations of pathogenic bacteria. Bacteriophages, also called phages, are viruses capable of infecting bacteria and loosely known as “bacteria eaters”. There are a few unique properties of bacteriophages over antibiotics, such as easy availability, diversity, autodosing (increase in number spontaneously), low inherent toxicity, specific host range, lack of cross-resistance with antibiotics, and low environmental impact. These properties are quite attractive when considering phages as alternatives to antibiotics. However, certain limitations of phage therapy include proper phage selection, narrow host range, effective formulation, probable immune reaction, and clinician understanding, which need to be addressed before its clinical application becomes a reality. While phage therapy is unlikely to be an absolute replacement of antibiotics, its application as an alternative therapy to topical infections has been successful where antibiotics have proved to be ineffective [88].

#### 9.11.2. Antivirulence Drugs 

A novel class of drugs called antivirulence drugs focuses on interfering with bacterial virulence factors instead of growth inhibition or killing of bacteria as an alternative approach to antibiotic therapy. Antivirulence drugs can disable specific bacterial proteins that are used by bacteria to attach themselves to host cells for initiation of infection. Thus, in disarming the bacteria, antivirulence drugs prevent the establishment of infection, and since they utilize a different mechanism of action, development of antibiotic resistance to antivirulence drugs is unlikely. Moreover, it has been shown that unlike conventional antibiotics, antivirulence drugs do not support drug-resistant bacterial strains to dominate over susceptible ones, and there is negligible perturbation of the healthy microbiota. The Food and Drug Administration (FDA) has approved the use of antivirulence drugs for bacterial-toxin-mediated diseases, and recently these drugs have been found to be effective against MRSA infections in mice. However, due to challenges in development and clinical use, antivirulence drugs would be suitable as adjunct or combination therapy with antibiotics [89]. 

#### 9.11.3. Bacteriocins 

Bacteriocins are natural antimicrobial peptides produced by certain bacteria that have bactericidal or bacteriostatic effects on similar or phylogenetically related bacterial strains. The harmless nature of bacteriocins would make them ideal agents, and a number of bacteriocins are now being studied for their potential use as antibacterial therapy. They are also being increasingly used to prevent the growth of dangerous bacteria in food, thereby extending the shelf life of food and delaying food spoilage. Nisin is an example of a bacteriocin derived from the lantibiotic family of antibacterial peptides, produced by certain gram-positive bacteria such as *Lactococcus* and *Streptococcus* species, and is widely used as a food preservative. Apart from its use in food production, nisin has now been found to have antibacterial activity against both gram-positive and gram-negative disease-associated pathogens including drug-resistant bacterial strains such as MRSA *Streptococcus pneumoniae*, *Enterococci, Clostridioides difficile*, and gram-negative bacteria such as members of enterobacterales [79].

### 9.12. One Health

The World Health Organization (WHO) was a partner in the 2008 establishment of a strategic “One Health” framework for approaching global health problems to achieve better public health outcomes in collaboration with the Food and Agriculture Organization of the United Nations (FAO) and the World Organization for Animal Health (OIE). It is an integrated collaborative, multisectoral, and transdisciplinary approach that is working at the local, regional, national, and global levels to attain optimal health for people, animals, and the environment, as defined by the One Health Initiative Task Force (OHITF). It focuses on a wide range of sustainable development objectives to design, implement, and monitor programs, policies, and research on AMR surveillance in order to substantiate evidence and to enhance advanced intersectoral collaboration between people, animals, plants, and their shared environment. Among the global health problems, AMR is the one that affects humans, the environment, and animals and most clearly illustrates the One Health approach [90].

The foundation of the One Health approach is based on three Cs: (1) communication, (2) coordination, and (3) collaboration between humans, animals, and environmental professionals to share their expertise in the One Health approach. Categories of people who may be involved in this approach are the three Ps: (1) pharmacists, (2) physicians, and (3) patients, along with other healthcare and epidemiology professionals. As far as the animal health representatives are concerned, veterinarians and agricultural workers are included, while ecologists and wildlife experts are represented as environmental experts [91].

The global assessment of the One Health approach and the tripartite commitment of the FAO–OIE–WHO group, would help AMR prevention through awareness programs, education about antibiotic usage, advocacy with political commitment, and antimicrobial stewardship. Meanwhile, novel computational and sequencing tools such as whole-genome sequencing (WGS) or next-generation sequencing (NGS) are advanced tools for studying AMR in different domains of One Health [92].

## 10. Conclusions

Evolution of antimicrobial resistance of bacteria is a continuous phenomenon occurring either by new chromosomal mutations or acquisition of drug-resistance genes through HGT. The incremental development of AMR over the previous two decades has created grave risk for global public health and is now appraised as the highest health danger in the 21st century, seriously limiting treatment options. MDR bacteria are frequently detected in many common infections such as respiratory, urinary, sexually transmitted, or tuberculosis infections globally. Meanwhile, development and supply of new antibiotics have lagged significantly since the 1980s and are not keeping pace with the speed of development of AMR. The future of successful antimicrobial therapy looks bleak in the context of unprecedented evolution of infections caused by multidrug-resistant pathogens and a paucity of the development of new antimicrobials. Unless global coordinated actions to stop the ongoing trend of AMR are adopted, a postantibiotic era for the 21st century can be a more real possibility than an apocalyptic fantasy.

Multiple drivers are contributing to the development and dissemination of antimicrobial resistance globally, creating a major concern for both human and animal health. Antimicrobial-resistant infections are more difficult to treat, leading to treatment failure and complications on top of huge financial costs to individuals and to the community. Prudent use of antibiotics with appropriate dosage and duration is among the most important means to reduce the selective pressure required for the emergence of resistant organisms. Strict practice of infection prevention and control measures in all healthcare facilities is a vital step in controlling the spread of MDR organisms [57,58].

Combating AMR requires an improved and coordinated global effort from all international governmental and nongovernmental agencies along with strong political momentum. Integration and the cooperation of policymakers, researchers, public health practitioners, pharma companies, hospital administrators, agriculture industry leaders, and members of the public are important in this endeavor. The unified and eventual goal of this collaboration is to decelerate the ongoing trends in AMR to minimize the health and economic burdens on people. Establishing antimicrobial stewardship and rigorous compliance of antibiotic policy in healthcare settings are invaluable strategies to combat antibiotic resistance. Further, good microbiology practice, surveillance, monitoring, minimizing OTC antibiotics and antibiotics in food animals, increasing access to quality and affordable medicines, vaccines, diagnostics, and enforcement of legislation are among the essential steps in mitigating the AMR problem [73,75,76,77,78].

Prevention is still the best strategy to reduce antimicrobial-resistant infections and their spread globally. While restoration of efficacy of existing antibiotics through their rational use is essential, urgent efforts should be exerted towards the development of new effective molecules of both antibiotics and alternatives to antibiotics, as well as new technological breakthroughs in diagnosis and vaccines development. Until now, many attempts have been made to address the problems of antibiotic resistance and the interventions required; however, coordinated action is largely absent, especially the political will at national and international levels. The forcible trend of antimicrobial-resistant infections indicates that within just a few years, we might face dire setbacks in the medical, social, and economical sectors, and all our achievements in modern medicine, such as major surgery, organ transplantation, treatment of preterm babies, and cancer chemotherapy, will vanish unless real and robust global coordinated actions are immediately taken [93].

## Figures and Tables

**Figure 1 healthcare-11-01946-f001:**
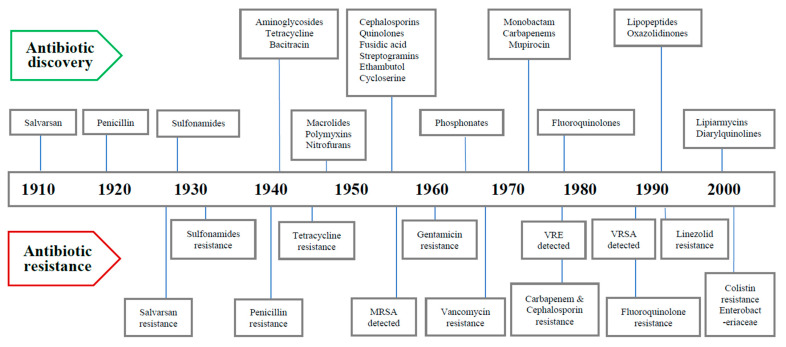
Timeline of discovery of major antibiotics and antibiotic resistance.

**Figure 2 healthcare-11-01946-f002:**
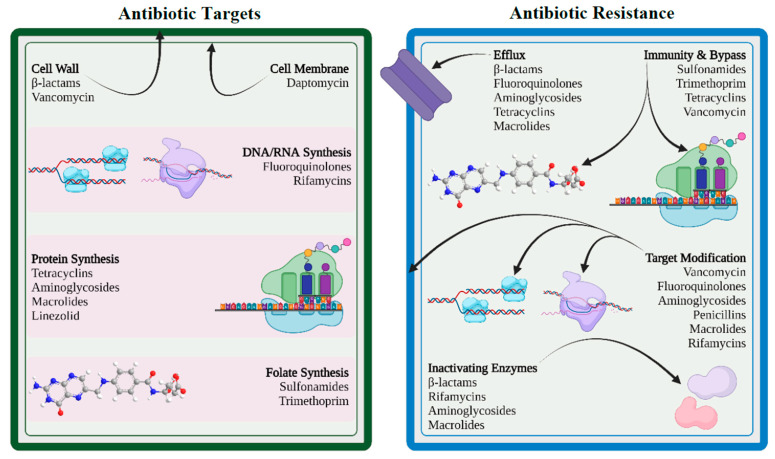
Antibiotic targets and mechanisms of drug resistance (created with BioRender.com (accessed on 12 November 2022)).

**Figure 3 healthcare-11-01946-f003:**
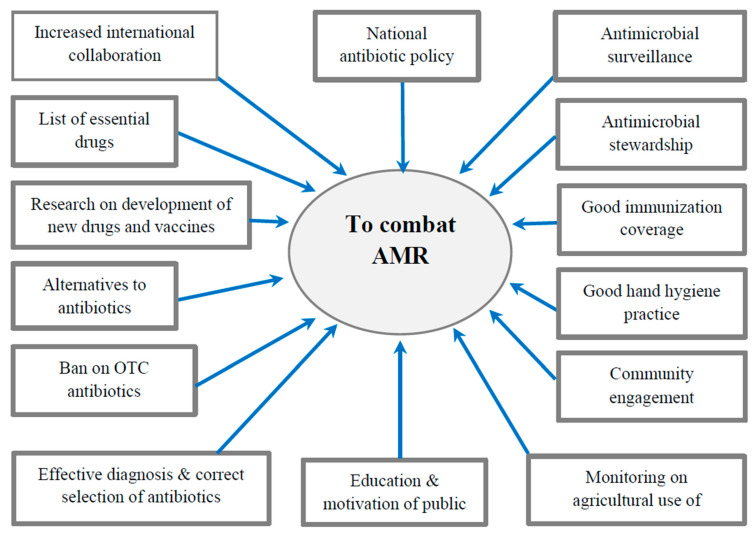
Major interventions to combat AMR.

## Data Availability

The data used to support this study are available from the corresponding author upon request.

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
