# Peer review of "Antimicrobial Resistance: A Growing Serious Threat for Global Public Health"

_healthcare, 2023, doi:10.3390/healthcare11131946_

Round 1
Reviewer 1 Report
Overall, this article provides a comprehensive overview of the evolution of AMR and strategies to combat AMR. Here're some of my suggestions:
1. I had trouble viewing some of the text in fig 1&3. Also, it would be better to change the titles from "Figure 1/3" to "Fig. 1/3" as that's how the authors referenced them in the text, and it would also be consistent with fig. 2
2. The introduction seems to jump around between different topics without a clear flow. For example, the author discusses antibiotic use in agriculture, then jumps to AMR, then back to different types of antimicrobials, then to antibiotic resistance specifically. Also, it seems that AMR is described in multiple places and ways. It would be beneficial to organize it in a more logical sequence.
Author Response
"Please see the attachment"

Reviewer 2 Report
The manuscript healthcare-2399687 is an interesting narrative review about the antimicrobial resistance.
The authors specify that they present a narrative review. However, the authors should provide a workflow about the method apply for the selection of the references / articles included in the manuscript.
More figures would add value to the manuscript. Also, the authors should provide more refences mostly at the Introduction part. There are big paragraphs with no references. When some write percentages, exact numbers, it must add the references (e.g. L90 - " Currently, 3.5% of active TB and 18% of previously treated TB cases belong to MDR-TB (multidrug-resistant tuberculosis)...".
In my opinion, for the clarity the authors could separate the historical data and the present / actual findings. It is only a personal opinion.
minor comments - the full-stop should be after the reference. - e.g. L 70 - ".[1-3]" should be "[1-3]."
- the latin names of the microorganisms should be in italics - L147 - "S. aureus", etc.
- the Enterobacteriaceae are change in Enterobacterales.
- please check the type=-error writing, minuscule / majuscule, the font, etc.
Author Response
"Please see the attachment"

Reviewer 3 Report
The antimicrobial resistance theme is a very relevant issue for the global public health.
Several differences in the type of text used are observed over the manuscript. This must be corrected. Also, lines 394-409 are all in italics, but different font size is present all over the manuscript.
All bacterial species or genus must be in italic, there are a lot of bacterial taxonomy problems in this manuscript
Several references to support the sentences are missing all over the manuscript, some examples are in the introduction section since line 50 to 68 there are no references to support the sentences and in lines74, 75, 77, 79, 82, 85,88, 93,95, 97, 99, 107, 109, 111, 112, 114, 155, 156, 185, 186, 193, 195, 197, 211, 217, 243,252, 254,272, 275, 277, 282, 283, 285, 286,290, 295, 351, 353,379, 389, 396, 401, 404, 405, 407, 412, 414, 425, 437, 439, 443, 446, 447, 452, 460, 463, and so on. Total review of the text with reference inclusion is crucial.
Then several errors are written, such as in line 93 “. Although antibiotics are invaluable in combating bacterial infections” or in line 96 “the unrestricted use of antimicrobials in livestock feed”.
In line 54 you mention that the antimicrobials are used in “agriculture as preventative measure for decades”. This is not correct. Also be careful with the indication of antimicrobials used as growth promoters, you should see the differences in countries because this is not applicable for example in Europe. You are saying this as a global use when several rules are applied all over the world.
In line 78 how did trade and finance influence antimicrobial resistance? I do not understand this statement.
In lines 208 -209 you say “Antibiotic resistance is an evolutionary response of bacteria when they are challenged with therapeutic antibiotics.” I do not agree with this sentence, the AMR can be potentiated by the misuse of antibiotics, however it is a natural phenomenon that always occurred even before the use of antibiotics. You only mention this in manuscript in line 387.
In lines 259- 260 you say “Animal feeds treated with antibiotics and their subsequent transfer to the humans through consumption of these animals constitute the direct route of acquisition of antimicrobial resistance from animals”, this statement is based on what? You should study the legislation applied to all animals products for human consumption and the antimicrobial residues verification in all these products.
In line 299 what do you mean by “ stop building of bactericidal concentrations over the entire biofilm”?
In line 382, what do you mena by fake drugs or substandard drugs?
The section 8 is a group of bullet points without any context.
The abstract and introduction section were very difficult to read. Please revise all manuscript.
Author Response
"Please see the attachment"

Reviewer 4 Report
AMR is a subject of interest to a wide audience and this review gives an overall description of several points related to AMR including alternatives to antibiotics.
I consider this manuscript suitable for publication but only after the authors address the following issues.
General comments: throughout the text we find some sentences or and words randomly written in italic with no reason and also with a higher letter size. Please correct. Moreover, I don´t understand why the references appear after the final dot of the sentences, is that a guideline of the journal?
Minor issues:
- Line 181: some words inside the boxes in figure 1 are not readable, the same happens in figure 3.
- Lines 394 to 409: why is the text in italic?
Author Response
"Please see the attachment"

Reviewer 5 Report
The manuscript entitled „Antimicrobial Resistance: A Growing Serious Threat for Global Public Health” gives a good overview about antibiotic resistance in general and on resistance mechanism level. Topic of this manuscript is an important issue however, some parts in the text should be revised.
Comments
1) The first sentence in the abstract should be revised. I suggest this form:
Antibiotics are among the most important discoveries of the 20th century, that have saved millions of lives from infectious diseases
2) You use „AMR” in many cases, but at the beginning of abstract you stated that „AMR” refers to „antimicrobial resistance”. Later in the abstract and in the manuscript you use this :
„The prevalence of AMR bacteria…”
„Therapeutic options of AMR bacterial infections..:”
In these sentences the „antibiotic resistant” form should be used. Please, check this in all over the manuscript.
3) Check in all over the manuscript that all bacterial names are in italic form „Staphylococcus aureus”, „S. aureus”, „Klebsiella pneumoniae”
4) Figure 1 should be adjusted to get to a better view on all details of this figure. All words should be properly visable!
5) Figure 1 should be extended, as in this form it shows the timeline from 1910 to 1960. But other novel antibiotics should be also shown in this figure.
6) „Clostridium difficile” is mentioned in the manuscript but it should be mentioned, that the current name is „Clostridioides difficile”
7) Use the terms uniformly. Both appear in text: „Gram positive” gram negative” .
I suggest these forms: Gram-positive; Gram-negative
8) Figure 3 should be adjusted to get to a better view on all details of this figure. All words should be properly visable!
9) In the reference list, both Ref 6 and Ref 61 are the same item. Please, modify!
Quality of English is good.
Author Response
"Please see the attachment"

Round 2
Reviewer 3 Report
Thank you for the responses provided.
However, I do not agree with the authors in line 93 “. Although antibiotics are invaluable in combating bacterial infections” - This is a completely wrong sentense; or in line 96 “the unrestricted use of antimicrobials in livestock feed”
Antibiotics are crucial in the treatment of bacterial infections since their discovery and use. Also there are restrictions in the use of antibiotics in animals, as well as antimicrobial residues control in animals food related products, so i consider that the authors must verify all this thematics all over the manuscript, and read a little more about them.
Also , you mention that the antimicrobials are used in “agriculture as preventative measure for decades” and did not correct this sentence. Antibiotics are used in animal production not in agriculture. This is a wrong statement.
Italics are still missing in the bacterial species descriptions in the manuscript (e.g. 148, 149, 150, 151, 154, 155, 156, 157, 168,
Missing references in entire paragraphs should not occurs ( e. g. lines 527-546)
The family names should not be in italics. Bacterial taxonomy still is a problem in this manuscript. Extensive revision is needed.
This manuscript is a review and not a scoping systematic review, so the inclusion of the paragraph in lines 121-128 is not needed.
Needs revision. Some sentences are difficult to understand. I recommend that a native person reads the article in order to verify this point.
Author Response
"Please see the attachment"
